# Dynamics of 2,4-D and Dicamba Applied to Corn Straw and Their Residual Action in Weeds

**DOI:** 10.3390/plants11202800

**Published:** 2022-10-21

**Authors:** Tiago Gazola, Renato Nunes Costa, Caio Antonio Carbonari, Edivaldo Domingues Velini

**Affiliations:** Department of Plant Protection, College of Agricultural Sciences, São Paulo State University (Universidade Estadual Paulista “Júlio de Mesquita Filho” UNESP), Botucatu 18610-034, SP, Brazil

**Keywords:** auxinic herbicides, diglycolamine salt, choline salt, leaching, monocotyledonous, eudicotyledonous, efficacy

## Abstract

2,4-D and dicamba are used in the postemergence management of eudicotyledonous weeds in different crops, most of which are grown under no-tillage systems. Due to the application methods for these products, their dynamics in straw and their residual action in soil have rarely been explored. Thus, the objective of this study was to evaluate the dynamics of 2,4-D and dicamba that have been applied to corn straw and to verify their relationship with residual control action in weeds. In the dynamics experiments, the herbicides were applied to 5 t ha^−1^ of straw, and rainfall simulations were performed with variable amounts and at different periods after application to evaluate herbicide movement in the straw. In the residual action experiments, the species *Digitaria insularis*, *Conyza* spp., *Bidens pilosa*, *Amaranthus hybridus*, *Euphorbia heterophylla*, and *Eleusine indica* were sown in trays, and 2,4-D and dicamba were applied directly to the soil, to the soil with the subsequent addition of the straw, and to the straw; all of these applications were followed by a simulation of 10 mm of rain. The physical effect of the straw and the efficacy of the herbicides in terms of pre-emergence control of the weed species were evaluated. The leaching of 2,4-D and dicamba from the corn straw increased with a higher volume of rainfall, and the longer the drought period was, the lower the final amount of herbicide that leached. The presence of the corn straw on the soil exerted a physical control effect on *Conyza* spp.; significantly reduced the infestation of *D. insularis*, *B. pilosa*, *A. hybridus*, and *E. indica*; and broadened the control spectrum of 2,4-D and dicamba, assisting in its residual action and ensuring high levels of control of the evaluated weeds. In the absence of the straw, 2,4-D effectively controlled the pre-emergence of *D. insularis*, *Conyza* spp., and *A. hybridus*, and dicamba effectively controlled *D. insularis*, *Conyza* spp., *B. pilosa*, *A. hybridus*, *E. heterophylla*, and *E. indica*.

## 1. Introduction

In the management of weeds, especially in the context of difficult-to-control species, the use of auxinic herbicides has become a valuable tool because, at a global scale, approximately 366 million ha are treated annually with these products, surpassed only by the herbicide inhibitors, acetolactate synthases (ALSs) (598 million ha) and 5-enolpyruvylshikimate-3-phosphates (EPSPs) (477 million ha) [1]. In Brazil, with the commercial releases of 2,4-D-resistant corn and soybean cultivars and dicamba-resistant soybean cultivars, through the technologies Enlist^®^ and Xtend^®^ [2], respectively, the use of these herbicides has increased.

These crops are produced mostly on previous crop remains and, therefore, a large part of these herbicide products is deposited on the remaining straw. The herbicides 2,4-D and dicamba are used exclusively in the postemergence control of eudicotyledonous weeds; however, these herbicides have important residual effects that are rarely explored, despite showing a reduction in the emergence of weed species such as *Convolvulus arvensis*, *Conyza* spp., *Chenopodium album,* and *Kochia scoparia* [3,4,5,6,7]. These residual effects on the soil optimize weed-management weeds by exerting direct action on the seed bank, allowing the crops to remain free of competition with the weeds for a longer period.

However, the straw present on the soil surface alters herbicide residual actions and behaviors, making them more susceptible to volatilization and/or photolysis [8]. The percentage of the herbicide that reaches the soil may also experience sorption or be leached, degraded, or absorbed by the plants present in the area [8,9]. All these processes alter the persistence of these products in the environment and determine the amount that will effectively be available to control the weeds. Evaluating the effect of dicamba applied in pre-emergence of the species *Bidens Bidens pilosa*, *Euphorbia hetrophylla,* and *Ipomoea nil*, Mundt et al. [10] observed that, in the systems that had the presence of straw, in the application and post add, and the application over the straw, approximately 60% of the weeds would be controlled if a concentration of dicamba in the soil is about 18 ng g soil^−1^.

These processes are related to the physicochemical properties of the herbicides, the amount and origin of the straw and the intensity and timing of the rain after the application of these products [11,12,13,14]. In this context, the herbicides 2,4-D and dicamba have physicochemical characteristics that may favor the leaching process, such as a low adsorption coefficient (Koc), low octanol-water partition coefficient, low persistence, and high water solubility (Table 1) [15].

In addition to the effect that the straw exerts on herbicide dynamics, its presence in the production system alters the population dynamics of the weeds by serving as a physical barrier that affects light and temperature and may also promote the release of compounds with allelopathic effects, in addition to containing animals and micro-organisms that can deteriorate the seeds [17,18,19,20,21]. These effects are related to the type, quantity, and uniformity of straw distribution; straw permanence duration and decomposition level in the soil; and, especially, the weeds that may or may not be affected by the presence of the straw [22]. Several studies have demonstrated the effect of the straw on weed suppression [21,23,24,25,26].

Given the above information, the objective of this study was to evaluate the dynamics of 2,4-D choline salt and dicamba salt of diglycolamine—DGA) in the corn straw and their relationship with their residual control action in the species *Digitaria insularis, Conyza* spp., *Bidens pilosa, Amaranthus hybridus, Euphorbia heterophylla*, and *Eleusine indica.*

## 2. Results

### 2.1. Dynamics of 2,4-D and Dicamba in Corn Straw

Rainfall volume and the period after rainfall directly affect the dynamics of 2,4-D and dicamba in corn straw. Thus, the adjusted model of Mitscherlich (Equation (1)) was adequate for understanding the movement process of herbicides in the straw because it showed a high coefficient of determination during all evaluation periods for both herbicides (Table 2). Parameter “a” of the model was the maximum amount of herbicides recovered for the total accumulated rainfall in each evaluated situation. Therefore, the final differences were observed in relation to the herbicide release in each evaluated period after a rainfall of up to 100 mm. Thus, the longer the period without rain, the lower the final amount of herbicide that was recovered (Table 2).

In comparison to dicamba, 2,4-D moved more through the corn straw. For 2,4-D, the occurrence of rain immediately after its application resulted in the greatest amount of the herbicide being transferred through the straw at 86.4% (834 g a.e. ha^−1^) in relation to the total amount of product applied after a rainfall higher than 50 mm (Table 2, Figure 1A,B). With the increase in days without rain, the movement of the herbicide through the straw was reduced. At 1, 3, 7, and 15 DAA, the maximum leaching was 76.4, 73.3, 66.7, and 51.7%, respectively, at rainfall volumes higher than 50 mm (Figure 1B). Regardless of the period without rain and, although the maximum leaching amount occurred with a rainfall volume higher than 50 mm, the herbicide dynamics with lower rainfall volumes were altered according to the dry season.

For 50% of the applied 2,4-D herbicide to move through the straw, 7 mm of rainfall was necessary, and 8, 10, 10, 17, and 39 mm were needed at 0, 1, 3, 7, and 15 DAA, respectively. Therefore, when the rainfall application occurred during periods of intense rainfall, the retention effects of the straw were not as pronounced compared to those when the rainfall application occurred during periods with lower rainfall amounts. Thus, the longer it takes for the first rain events to occur, the higher the amount of water needed for herbicide movement to occur and the lower the final amount of 2,4-D that will reach the soil.

The movement of dicamba through the corn straw was similar to that of 2,4-D in terms of rainfall volume and the dry periods after the application but to a lesser extent. The occurrence of rain immediately after the dicamba application caused the maximum transfer of the herbicide through the straw at 61% (284.7 g a.e. ha^−1^) in relation to the total amount of product applied after a rainfall higher than 27 mm (Table 2, Figure 1C,D). With the increase in days without rain, herbicide movement in the straw was reduced; at 1, 3, 7, and 15 DAA, the maximum leaching was 54.9, 54.7, 47.5, and 44.6%, with rainfall volumes higher than 34, 38, 56, and 52 mm, respectively (Figure 1D). As in the 2,4-D dynamics, the volume of rainfall required for higher product retention in the straw will be higher as the dry period increases after the herbicide application. Therefore, even with the intense rainfall after the application of dicamba, the movement in the straw was still high, and a lower amount of dicamba reached the soil compared to that of 2,4-D.

### 2.2. Effect of Straw and Residual Action of 2,4-D Choline Salt and Dicamba DGA Salt on Weeds

When analyzing the physical effect of the straw on weed emergence, it was observed that, for *E. heterophylla,* there was no reduction in the number of plants and biomass accumulation, while *E. indica, D. insularis,* and *Conyza* spp. were sensitive, with 83, 90, and 100% controlled, respectively. For *B. pilosa* and *A. hybridus*, the presence of the straw promoted control up to 60% (Figure 2).

By highlighting the physical effect of the straw on the weeds (Figure 2), when integrating 2,4-D and dicamba in the system, these herbicides interfered with weed control during pre-emergence (Table 3; Figure 3 and Figure 4). The herbicides applied to the soil, either by direct application or by movement in the straw (Figure 1), influence the emergence of mono- and eudicotyledonous weeds (Figure 3 and Figure 4). Therefore, when a production system contains straw, the dynamics of the herbicides are altered, which is reflected in the emergence of weeds.

For *D. insularis*, when 2,4-D was applied to the soil with subsequent straw cover followed by a simulation of 10 mm of rain (AS + S) or when the application occurred directly on the corn straw followed by a simulation of 10 mm of rainfall (ASC), only 0.7 g a.e. ha^−1^ of the herbicide was sufficient to promote 100% control (Figure 3). When dicamba was applied in the AS + S and ASC systems at only 0.4 and 0.7 g and ha^−1^, the same 100% control was obtained, respectively (Figure 4). Notably, only the physical effect of the straw caused 90% control in *D. insularis* (Figure 2).

For *B. pilosa*, the application of 2,4-D in the AS + S system at doses of 56.48 and 113.25 g a.e. ha^−1^ was sufficient to promote 80 and 90% control, respectively, and 100% was obtained from 1616 g a.e. ha^−1^ (Figure 3). In the ASC system, these same control levels were obtained with doses of 117.4 g a.e. ha^−1^ (80%) and 345.7 g a.e. ha^−1^ (90%), and the maximum control obtained was 98% from 2436 g a.e. ha^−1^ (Figure 3). With the application of dicamba in AS + S, only 0.03 and 0.05 g a.e. ha^−1^ were sufficient to promote 80 and 90% control, respectively (Figure 4). For the application of dicamba in the ASC system, doses of 40.3 and 71.6 g a.e. ha^−1^ were necessary to obtain the same 80 and 90% control, respectively. For *B. pilosa*, the isolated effect of the straw generated a control level at almost 60% (Figure 2); thus, herbicide movement within the straw was verified, even if limited (Figure 1), and with the application of low doses of dicamba, it was possible to increase the control level to 100% with a dose of 0.3 g a.e. ha^−1^ in the AS + S system and a dose of 646 g a.e. ha^−1^ in the ASC system (Figure 4).

As with *B. pilosa*, the physical effect of the straw also provided control at almost 60% for *A. hybridus* (Figure 2); however, when 2,4-D was applied in the AS + S system, a dose of 0.3 g a.e. ha^−1^ was sufficient to promote 100% control (Figure 3). Conversely, with the application of 2,4-D in the AS + S system, 1031 g a.e. ha^−1^ was needed to obtain the same 100% control (Figure 3). In the application of dicamba in the AS + S system, 0.3 g a.e. ha^−1^ was sufficient to promote 100% control, while in the ASC system, 503 g a.e. ha^−1^ was needed (Figure 4).

In addition to the physical effect of the straw not interfering with the control of *E. heterophylla* (Figure 2), this species was also the least sensitive to the action of 2,4-D. In its pre-emergence, 1959.2 and 3709.9 g a.e. ha^−1^ were needed to obtain 80 and 90% control, respectively (Table 3, Figure 3). The application of 2,4-D in the systems with the straw also did not result in high levels of control since, when the application was in the ASC system, the maximum control (67%) was obtained at high doses (from 3,485 g a.e. ha^−1^) (Figure 3). Thus, at commercial doses (maximum of 1368 g a.e. ha^−1^), the effect of 2,4-D on the control of *E. heterophylla* in a conventional or no-tillage system would be limited. Conversely, dicamba was effective in controlling *E. heterophylla*, even in the worst scenario, where the herbicide had to move through the straw (ASC), doses of 422.6 and 794.1 g a.e. ha^−1^ were sufficient to obtain 80 and 90% control, respectively (Figure 4).

Unlike *D. insularis*, the monocotyledonous *E. indica* was not well controlled by the action of 2,4-D during its pre-emergence in the absence of the straw. However, because the physical effect of the straw provided control close to 80%, when 2,4-D was applied in the AS + S and ASC systems, 54.2 and 161.3 g a.e. ha^−1^ were sufficient to promote 90% control, respectively (Table 3). This control reached 100% in the AS + S system with 413.5 g a.e. ha^−1^ and 99% in the ASC system with 1205 g a.e. ha^−1^ (Figure 3), demonstrating that 2,4-D moved through the corn straw (Figure 1) and contributed to the control of *E. indica*. The species were more sensitive to the application of dicamba than that of 2,4-D, and the presence of the straw did not impair the action of the herbicide since, at doses of 425.5 g a.e. ha^−1^ in the AS + S system and the ASC system, 100% control and 98% control occurred, respectively (Figure 4). Therefore, for the applications with commercial doses of dicamba (720 g a.e. ha^−1^) in a no-tillage production system, where the herbicide has contact with the straw and with rainfall that enables the herbicide to move through the soil (Figure 1), the control of this species will be highly effective.

With the application of 2,4-D in the AS system, it was observed that the species *D. insularis*, *Conyza* spp., and *A. hybridus* were more sensitive to the action of the herbicide because, at doses close to 755, 323, and 269 g a.e. ha^−1^, respectively, 90% control was obtained (Table 2; Figure 3). For *B. pilosa* and *E. heterophylla* to obtain the same level of control, the required dose was 2055 and 3710 g a.e. ha^−1^, respectively. For *E. indica*, it was not possible to obtain 90% control within the range of doses studied, and its control required a theoretical dose of 8702 g a.e. ha^−1^ (Table 2; Figure 3). For dicamba in the AS system, all the species were sensitive to the application of the herbicide at pre-emergence, with 90% control at doses of 131.9, 94.4, 135.6, 67.1, 172.5, and 610.5 g a.e. ha^−1^ for *D. insularis, Conyza* spp., *B. pilosa, A. hybridus, E. heterophylla*, and *E. indica*, respectively (Table 2; Figure 4).

In general, for all the species evaluated, when the isolated effect of the straw did not provide satisfactory control (Figure 2), higher concentrations of 2,4-D and dicamba needed to be applied in the ASC system than in the AS + S to obtain the same control effects (Figure 3 and Figure 4). This occurred because in the ASC system, the herbicides had to leach through the corn straw to reach the soil solution. When analyzing the herbicide dynamics in the straw, with a simulation of 10 mm of rain immediately after the commercial herbicide application (same volume of rain used in this efficacy experiment) doses, only 575.82 g a.e. ha^−1^ (59.07%) of 2,4-D and 281.34 g a.e. ha^−1^ (48.65%) of dicamba was transferred to the vegetation cover (Figure 1). However, this decreased leaching amount does not imply adjustments to product doses because, in all cases where vegetation cover was present, with the exception of *E. heterophylla*, these concentrations were sufficient to exert an over 90% control effect for the species (Figure 3 and Figure 4).

Thus, an important point to be considered is the need for rainfall after the application of 2,4-D and dicamba during weed pre-emergence so that the herbicides can reach the soil solution and control the weeds. Normally, when desiccation occurs when other postemergent herbicides are applied, rain needs to have already occurred before the physiological conditions of the weeds are adequate so as not to affect the action of the products. However, this is a positive factor that makes the pre-emergence position more flexible because the occurrence of rainfall immediately after the application will not affect the control action, as the focus is not on the plants present but on the residual effect of the herbicides on the seeds of the species present in the soil.

## 3. Discussion

Several authors have highlighted the importance of rainfall intensity and timing after the herbicide application in relation to the leaching process in the straw and how vegetation cover interferes with this process [11,12,13,14,27,28]. A study showed that, when sulfentrazone is applied to sugarcane straw in periods without rain, there are significant losses in terms of the ability of the herbicide to reach the soil [14]. The same scenario occurs for tebuthiuron, where small amounts of straw can intercept almost the entire spray solution; thus, rainfall needs to occur soon after the application since the loading of the herbicide into the soil is reduced with the increase in the interval between the application and the first rain [27].

As previously noted, because these products are recommended for use exclusively during postemergence, the dynamics of the herbicides 2,4-D and dicamba in the straw are not addressed in the literature, although they have residual action in the soils. Therefore, the results obtained in these experiments address this gap in knowledge and are important given the increased use of these herbicides and the release of cultivars resistant to them [29,30,31]. In addition, no-tillage systems require even higher attention because the dynamics of these herbicides will be altered due to the presence of the straw. In Brazil, the production of corn prior to soybean cultivation is common and, consequently, a significant amount of the herbicides applied in this system will be deposited on the corn straw. Worldwide, more than 120 million ha are cultivated under no-tillage systems and, in Brazil, the cultivated area of cereals alone exceeds 33 million ha [32].

The availability of an herbicide in the soil, when applied during pre-emergence, is related to its physicochemical characteristics and movement, vegetation characteristics, and the rainfall regime that occurs after its application [33,34,35]. Another important factor that explains the decrease in herbicide leaching in the straw is the degradation of the product, which can occur mainly due to photolysis and/or volatilization [9].

For photodegradation of an herbicide molecule to occur, its maximum absorbance must occur at wavelengths in the range of 290 to 400 nm because wavelengths below 290 nm are absorbed by the ozone layer and do not reach the Earth’s surface, and those larger than 400 nm do not have enough energy to promote the breakdown of molecules [36,37,38]. 2,4-D and dicamba are not molecules that undergo photodegradation under environmental conditions; reports in the literature indicate that the maximum light absorbance occurs at wavelengths below 290 nm [15,39,40]. Thus, the decrease in herbicide leaching levels reported in this study was not due to photodegradation.

For volatilization to occur, the vapor pressure and solubility of the herbicide, as well as the temperature and humidity conditions, vegetation cover, dead vegetation cover, and the treated soil, influence this process. 2,4-D ester formulations are considered to have high volatility and amine formulations are considered to have low volatility [41]. The most current formulation, choline salt, was used in these experiments and is considered even less volatile than the amine formulations [42]. Dicamba is a weak acid with a pKa of 1.87 [15] and its dissociation can have a substantial impact on volatility since its acid form is the main form that is volatilized [31,43,44].

Studies evaluating the response of cotton to 2,4-D volatilization for the amine and choline formulations have shown that cotton experienced injury at a rate less than 2% with the amine formulation, and there were no cotton injuries with the choline formulation [42]. In an environment more prone to volatilization, injury levels of 76, 13, and 5% were observed in cotton exposed to 2,4-D in the ester, amine, and choline formulations, respectively [45]. Studies indicate that the volatilization of dicamba in the formulation of diglycolamine salt—DGA (same used in these experiments), is 50% lower compared to that in the formulation of dimethylamine salt—DMA [43,46,47], and the new formulation of DGA containing acetic acid further reduces this volatilization because it inhibits the dissociation and formation of dicamba acid [48,49]. Thus, the volatility of dicamba can be managed through additives in mixtures and the use of safer formulations, both in terms of dicamba and other products used in the mixture [50].

The deposition surface also influences the dicamba volatilization rate. The application of the DMA formulation during the postemergence of soybean and corn increases the volatilization by 35% when compared to its application directly to the soil [51]. When DGA dicamba is applied directly to the corn straw, volatilization (approximately 3.5 ng cm^−2^) is even lower than that in wet soil (9.11 ng cm^−2^) [52]. This is because dicamba has high solubility and low affinity with soil colloids and is more likely to be volatilized [53]. The DGA formulation without acetic acid (the same used in these experiments) showed low levels of volatilization in the corn straw and dry soil [52].

Therefore, it is unlikely that the volatilization levels of 2,4-D choline and dicamba salt DGA, which may have occurred in the corn straw, caused the reduction in the leaching rate observed in these experiments, although it may have contributed to the process (Figure 1). Given these facts, the most plausible explanation for the reduced leaching rate is that the amount of 2,4-D and dicamba that did not leach from the corn straw may have been retained in it. However, the higher the water solubility is, the higher the capacity of an herbicide to move through the vegetation cover and reach the soil [54], and the herbicides 2,4-D and dicamba have high water solubility (Table 1). Nevertheless, with 100 mm of rain at 15 DAA, only 51.7 and 44.6% of 2,4-D and dicamba, respectively, moved through the corn straw (Figure 1). One possible explanation is that during the straw-ageing process, the lignin and cellulose in the cell walls degrade; this process exposes the herbicide loads present in the straw and, consequently, increases the retention of the herbicide due to the higher contact surface [55,56].

The persistence of 2,4-D and dicamba in the corn straw can benefit weed management, prolonging the residual control effect on the weeds as a function of their gradual release into the soil as rainfall occurs. However, the period without rain after the application of these herbicides cannot be very long because, the longer it is, the lower the final leached concentration will be. A lower leach concentration could directly affect weed control efficacy (Figure 3 and Figure 4) as the available concentrations would not be sufficient to control the plant species since, even with large rainfall volumes, there would be no effective increase in the leachate product (Figure 1).

Under field conditions, Mundt et al. [10] evaluated the dynamics of dicamba in different application systems and it was observed that the higher potential for dicamba leaching was applied directly to the soil. For the application over the corn straw, the maximum concentration of dicamba in the soil was found after 10 mm of rain. Dicamba was not found in the different soil layers after 40 mm of rain for the soil and straw application.

Carbonari et al. [14] evaluated sulfentrazone applied at a dose of 750 g a.e. ha^−1^ on 10 t ha^−1^ of sugarcane straw and simulated an accumulation of 100 mm of rain at 1, 30, and 60 DAA, followed by a new simulation of 20 mm of rain at 7 and 14 days after the occurrence of the first 100 mm. They found that the maximum amount of the herbicide that was released from the straw was only 0.05%. Thus, 100 mm of accumulated rainfall seemed to promote the maximum possible release of the herbicides applied to the straw, and it is unlikely that an additional release would have occurred due to subsequent rainfall.

For 2,4-D, with the occurrence of rain immediately after its application (0 to 3 DAA) at 20 mm, 90% of the total product leached, relative to an accumulated rainfall of 100 mm, and 99% leached starting at 40 mm (Figure 1). When there are longer periods without rain (15 DAA), 90 and 99% of the maximum leachate will occur at 27 and 53 mm of rain, respectively (Figure 1). For dicamba, in rain simulations performed immediately after its application and at 1 and 3 DAA to obtain a level of 90% of the total leached at 100 mm rainfall, it would be necessary to have 15-, 18-, and 20-mm rain events, respectively, and, for longer periods without rain (7 and 15 DAA), 29 mm would be necessary (Figure 1). To reach 99%, at 0, 1, and 3 DAA, the rainfall volume would need to be 30, 38, and 40 mm, respectively, and 56 mm in situations of 7 and 15 days without rainfall (Figure 1). Therefore, for 2,4-D and dicamba, their dynamics in the straw as a function of the rainfall regime should be taken into account in the spraying of these herbicides, especially when aiming to ensure their residual action in the weeds, as this factor directly affects the concentration available for the soils.

Experiments that evaluate the residual effect of 2,4-D and dicamba on weeds are extremely rare. The only reports found in the literature describe that 2,4-D applied repeatedly, whenever the conditions for emergence of *Convolvulus arvensis* plants were favorable, was able to significantly reduce its emergence [3]. Johnson et al. [7] reported the efficacy of dicamba (273 g a.e. ha^−1^) in the pre-emergence of *Conyza* spp. and *Chenopodium album*. Recently, Ou et al. [6] observed that dicamba (560 g a.e. ha^−1^) controlled *Kochia scoparia* pre-emergence, provided that the infestation of resistant biotypes did not exceed 600 viable seeds m^2^.

Thus, the results presented in this study showed that 2,4-D and dicamba have an important residual action in the weeds, including contributing to the management of monocotyledonous species, which does not occur when they are used postemergence. This information is of great importance due to the high infestation of *D. insularis* and *E. indica* that occurs in agricultural regions of Brazil, mainly due to the multiple resistance to EPSPS and ACCase-inhibiting herbicides [57] and the lack of new products that efficiently control them postemergence.

The herbicides 2,4-D and dicamba, when applied postemergence, act exclusively on eudicotyledons because they do not move through grasses well and because they are able to metabolize them quickly into nonphytotoxic compounds (2,5-dichloro-4-hydroxyphenoxyacetic acid; 2,3-dichloro-4-hydroxyphenoxyacetic acid; and 3,6-dichloro- *o*-anisic acid (5-OH dicamba)) irreversibly [58,59,60]. The high efficacy of the studied herbicides during the pre-emergence of monocotyledons may be related to the absence or limitation of the metabolization of the herbicides since their residual action occurs in the seeds during the germination process, preventing the emergence of the seedlings.

Another possible explanation for the action on monocotyledons is that both 2,4-D and dicamba may have interrupted cell division and/or increased the production of abscisic acid (ABA), preventing the digested reserves in the seeds from being assimilated for the formation of new tissues, protrusion of the primary root, and seedling formation. In the germination process, with high levels of auxin, there is no expression of the genes necessary for the cell to evolve from protein synthesis and growth to the DNA duplication phase [61,62], in addition to activating genes encoding the proteins responsible for ethylene biosynthesis and ABA overproduction [63,64,65]. ABA inhibits germination by repressing the synthesis of hydrolytic enzymes that are essential for breaking down the seed reserves during the process [66].

## 4. Materials and Methods

The experiments were conducted between 2019 and 2020 in greenhouses and laboratories at the Nucleus for Advanced Research in Matology—NUPAM, College of Agricultural Sciences, São Paulo State University (Universidade Estadual Paulista “Júlio de Mesquita Filho” UNESP), Botucatu, Sao Paulo, Brazil (22°50′35.60″ S and 48°25′29.00″ W).

### 4.1. Dynamics of 2,4-D Choline and DGA Salt in Corn Straw

To evaluate the dynamics of the herbicides 2,4-D and dicamba on the corn straw, two experiments were conducted for each herbicide in a completely randomized factorial design of 6 (rain blades) × 5 (days without rain), with 5 replicates. The treatments consisted of rainfall event simulations with accumulated amounts of 5, 10, 20, 35, 50, and 100 mm at 0, 1, 3, 7, and 15 days after application (DAA), respectively, of the herbicides 2,4-D choline salt at a concentration of 912 g a.e. ha^−1^ (Enlist^®^ Colex -D—456 g a.e. L^−1^) and dicamba salt diglycolamine—DGA at a concentration of 480 g a.e. ha^−1^ (Atectra^®^—480 g a.e. L^−1^).

The experimental units corresponded to 4.5 cm diameter containers (polypropylene laminate capsules) filled with straw and coupled to collection tubes. The straw was obtained from the leaves of corn plants at the senescence stage, collected from crops with no history of herbicide application, and the straw was cut into small fragments of approximately 0.5 × 1.0 cm and weighed on a precision scale of 0.0001 g in an equivalent amount at 5 t ha^−1^. Extra containers with glass slides, both in the same measurements described above, were used to quantify the total amount of each herbicide deposited.

For the herbicide applications, an automated sprayer was used in a closed environment with a 2-m wide spray boom equipped with four XR 11002 VS nozzles spaced 0.5 m apart from each other and placed at a height of 0.5 m relative to the experimental units. The working pressure used by the equipment was 2.0 kgf cm^−2^, with a velocity of 3.6 km h^−1^ and a spray application of 200 L ha^−1^. At the time of the application, the temperatures were 25 and 29 °C, and the relative humidity was 60 and 53% for tests 1 and 2 of each herbicide, respectively.

Immediately after the application, the targets that did not contain straw were washed with 40 mL of deionized water, and aliquots were filtered in a Millex HV filter (Millipore) 0.45 µm with a 13 mm Durapore membrane and stored in a 9 mm amber vial (Flow Supply) with 2 mL of capacity for subsequent chromatographic analysis.

The experimental units were kept in an external environment under solar radiation and protected by a structure with a quartz cover until the specific rain simulation periods occurred. The quartz cover was used to allow the passage of ultraviolet wavelengths (290 and 400 nm, which are the bands responsible for promoting the photodegradation of herbicides) through the units to the surface [38].

To simulate rain events, a second spray boom was coupled to the same sprayer described above on the experimental units. The boom consists of 10 SprayJet^®^ DG 9505 EVS spray nozzles spaced 6.5 cm apart and positioned 0.5 m above the target. The system was calibrated with a working pressure of 2.0 kgf cm^−2^, allowing the production of variable rain events so that, at each movement of the application bar, a cumulative 2.5 mm of rainfall was generated.

To evaluate the dynamics occurring within the straw, for each period after the herbicide application (0, 1, 3, 7, and 15 DAA), and for each accumulated rainfall amount (5, 10, 20, 35, 50, and 100 mm), the solutions resulting from the movement within the straw were measured, and aliquots were filtered through a Millex HV filter (Millipore) 0.45 µm, with a 13 mm Durapore membrane and stored in a 9 mm amber vial (Flow Supply) with 2 mL capacity for subsequent chromatographic analysis.

The herbicides were analyzed using a liquid chromatography tandem mass spectrometry (LC-MS/MS) system consisting of a high-performance liquid chromatograph (HPLC) (Shimadzu, Proeminence UFLC, Kyoto, Japan) equipped with two LC-20AD pumps, a SIL-20AC autoinjector, a degasser DGU-20A5, a CBM-20A controller system, and a CTO-20AC oven. Coupled to the HPLC, a Triple Quad 4500 mass spectrometer (Applied Biosystems, Foster City, CA, USA) was used.

The 2,4-D chromatographic analyses were conducted with a C18 column (Synergi 2.5 μ Hydro RP 100 Å) using an injection volume of 20 μL, with 0.1% formic acid in water and 0.1% formic acid in methanol. The flow rate used was 0.6 mL min^−1^, and the proportion of solvents gradually increased from 40:60 (methanol/water) from 0 to 1 min to 95:5 from 1 to 6 min and returned to the initial conditions from 6 to 10 min. The total run time was 10 min, with a retention time of 4.18 min in the chromatographic column.

The chromatographic analyses of dicamba were conducted with a C18 column (Kinetex 2.6 μM phenyl-hexyl 100 × 2.1 MM) in an injection volume of 20 μL, with 0.1% formic acid in water and 0.1% formic acid in methanol. The flow rate used was 0.3 mL min^−1^, and the proportion of solvents gradually increased from 25:75 (methanol/water) from 0 to 2 min to 95:5 from 2 to 8 min and returned to the initial conditions from 8 to 15 min. The total run time was 15 min, with a retention time of 5.36 min in the chromatographic column.

The electrospray ionization source (EIS) was used in the negative and positive modes. Eight concentrations of the 2,4-D and dicamba analytical standards with certified purity levels of 99.8% and 99.9% (Sigma Aldrich, St. Louis, MO, USA), respectively, were included in the calibration curve.

### 4.2. Residual Action of 2,4-D Choline and Dicamba DGA Salt in Weeds

Two experiments were conducted for each herbicide in a completely randomized design composed of 8 doses and 4 replicates to determine the efficacy of the herbicides controlling the pre-emergence of the weeds *D. insularis*, *Conyza* spp., *B. pilosa*, *A. hybridus*, *E. heterophylla,* and *E. indica* in the absence and presence of the straw on the soil, both followed by rainfall simulations.

The number of seeds of each species was standardized by weight, with the seed weight of *D. insularis* being 0.200 g, that of *Conyza* spp. being 0.400 g, that of *B. pilosa* being 0.350 g, that of *A. hybridus* being 0.050 g, that of *E. heterophylla* being 0.400 g, and that of *E. indica* being 0.070 g. After weighing, the seeds of all the species were mixed for subsequent sowing in trays (27 × 41 cm) with a capacity of 7.5 L. The soil used had the following physicochemical characteristics: 281, 87, and 632 g.dm^−3^ of clay, silt, and sand, respectively; pH (CaCl_2_) = 5.3; OM = 11 g dm^−3^; P (resin) = 18 mg dm^−3^; Al^3+^ = 2 mmolc dm^−3^; H + Al = 40 mmolc dm^−3^; K^+^ = 0.8 mmolc dm^−3^; Ca^2+^ = 21 mmolc dm^−3^; Mg^2+^ = 9 mmolc dm^−3^; SB = 31 mmolc dm^−3^; cation exchange capacity (CEC) (T) = 70 mmolc dm^−3^; S = 56 mg dm^−3^; and V% = 42.

For sowing, the trays were filled with 3.5 L of dry and sieved soil, and a metal frame (24 × 38 cm) was placed in their center with the seeds evenly distributed inside. Subsequently, the soil was covered with an additional 1.5 L of soil, and the frame was carefully removed so that the seeds were arranged only in the useful area at approximately 1 cm of depth. This procedure was necessary to ensure that the physical effect of the straw on the emergence of the seedlings was not impaired and that the seeds were not in contact with the edges of the trays. The straw used was the same as that in the previous tests; however, the fragments had dimensions of approximately 5 × 3 cm, but the proportion of 5 t ha^−1^ remained.

The treatments consisted of applications of the herbicide 2,4-D choline salt at doses of 0, 57, 114, 228, 456, 912, 1824, and 3648 g a.e. ha^−1^ and dicamba salt DGA at doses of 0, 30, 60, 120, 240, 480, 960, and 1920 g a.e. ha^−1^ followed by a simulation of a 10 mm rain event and the herbicide application as a function of the straw, where AS is the application of the herbicide directly on the sown soil followed by a rainfall simulation; AS + S is the application of the herbicide directly on the sown soil followed by a straw cover and a rainfall simulation; and ASC is the application of the herbicide directly on the straw covering on the sown soil followed by a rainfall simulation.

During the experiments, minimal water replenishment was performed according to the plants’ needs. To ensure the simulated rainfall volume was standardized, in the treatments with the presence of the straw, water was carefully replenished with the aid of a pick at the edges of the trays. This method was used to prevent new straw surface wetting, thus ensuring the adopted rainfall simulations were standardized. In the treatment without the straw, wetting also occurred with the picks but evenly throughout the useful area of the tray.

Herbicide spray applications and rain simulations were conducted with the same spray simulator and the same configurations described in the previous experiment. The climatic conditions at the time of the application were temperatures of 26 and 28 °C and relative humidities of 61% and 55% for tests 1 and 2 of each herbicide, respectively.

To evaluate the effect of the straw with the use of 2,4-D and dicamba, we first sought to understand the isolated effect of the corn straw on the emergence of the weeds. For this purpose, we collected data when no herbicides were applied and analyzed the emergence and biomass of the weeds in the soil with and without the presence of the straw. The plots were subjected to an evaluation of the control percentage at 21 DAA by counting the number of emerged plants and shoot dry mass for each species.

### 4.3. Data Analysis

As all experiments were conducted in duplicate, the data obtained were subjected to analysis of variance independently, and the homogeneity test of residual variances (Fmax) was applied [67]. Once the homogeneity between the experiments was confirmed, the joint analysis was continued, and a new analysis of variance was performed, in which the data were grouped and considered as an experiment with 10 replicates for each herbicide in the dynamics assay and with 8 repetitions for the residual action test.

The results of the straw transposition were subjected to analysis of variance by the F test, and the confidence interval (*p* ≤ 0.05) was calculated for the values expressed in g a.e. ha^−1^, represented by the error bars. If significant, then the herbicide concentration data present in the solutions were converted into percentages relative to the total herbicide applied and adjusted to the nonlinear regression model developed by Mitscherlich [68] (Equation (1)):(1)y=a [1−10(−c(x+b))]
where a, b, and c correspond to the parameters of the equation; parameter a represents the maximum asymptote of the curve, which indicates the maximum herbicide recovered for the total accumulated rainfall; parameter b corresponds to the lateral displacement of the curve; and parameter c indicates the concavity of the curve. The value of y represents the total amount of herbicide extracted from the straw and x represents the rainfall (mm) required for the movement of the herbicide in the straw.

The results for the number of plants were subjected to analysis of variance by the F test. Being significant, they were converted into percentages in relation to the nonapplied controls and adjusted to the nonlinear regression model proposed by Streibig [69] (Equation (2)):(2)$$y=a/[1+{\left(x/x_{0}\right)}^{b}]$$
where y is the control percentage; x is the concentration of the herbicide in g a.e. ha^−1^; a is the amplitude between the maximum and minimum points of the variable; x_0_ is the concentration of the herbicide that provides a 50% response of the variable; and b is the slope of the curve around x_0_.

The dose-response curves were constructed from the equations. Based on the adjusted models, the concentrations of the herbicides that would provide 80 and 90% weed control (LC_80_ and LC_90_) were calculated. To perform the calculations, the following equation was used (Equation (3)):(3)$${\mathrm{LC}}_{F}={\left(F/100-F\right)}^{1/b}.x_{0}$$
where F indicates the LC to be calculated (LC_80_ or LC_90_) and b and x_0_ are the parameters of the Streibig equation [69].

The regression analyses were performed using the statistical software SAS^®^ (Statistical Analysis System, SAS Institute, version 9.1.3, Cary, NC, USA), and the plots were plotted in SigmaPlot^®^ (version 12.5).

## 5. Conclusions

The persistence of 2,4-D choline salt and dicamba salt DGA in the corn straw was high, and its leaching increased due to the rainfall period and volume.

The presence of 5 t ha^−1^ of corn straw on the soil exerted a physical control effect on *Conyza* spp. and reduced the infestation of *D. insularis*, *B. pilosa*, *A. hybridus*, and *E. indica* but did not interfere with the emergence of *E. heterophylla*.

In the absence of the straw, commercial doses of 2,4-D choline salt are effective in controlling the pre-emergence of *D. insularis*, *Conyza* spp., and *A. hybridus.* Under these same conditions, dicamba salt DGA is effective in controlling *D. insularis*, *Conyza* spp., *B. pilosa*, *A. hybridus*, *E. heterophylla*, and *E. indica*.

The presence of the corn straw on the soil broadened the control spectrum of 2,4-D and dicamba, and improved the control efficacy of the weeds studied, except for *E. heterophylla*.

## Figures and Tables

**Figure 1 plants-11-02800-f001:**
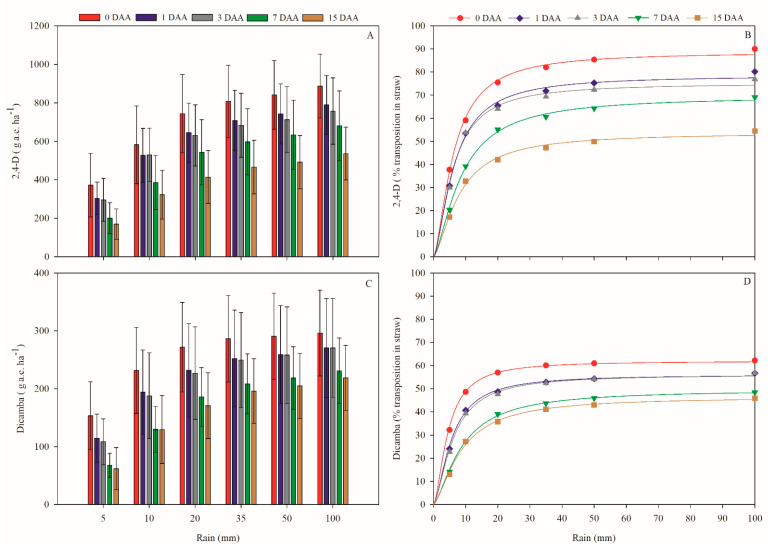
Dynamics of 2,4-D and dicamba in corn straw at g a.e. ha^−1^ (**A**,**C**) and percentage (**B**,**D**) of herbicides leached into the straw in relation to the total amount deposited after rainfall simulations (5, 10, 20, 35, 50, and 100 mm) in different periods after herbicide application (0, 1, 3, 7, and 15 days after application—DAA). The error bars represent the confidence interval (*p* ≤ 0.05).

**Figure 2 plants-11-02800-f002:**
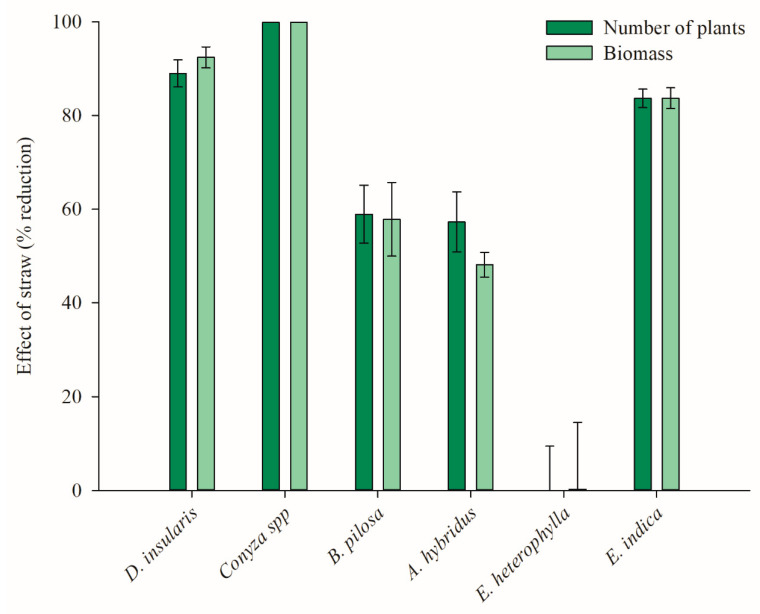
Effect of corn straw on weed emergence and biomass accumulation.

**Figure 3 plants-11-02800-f003:**
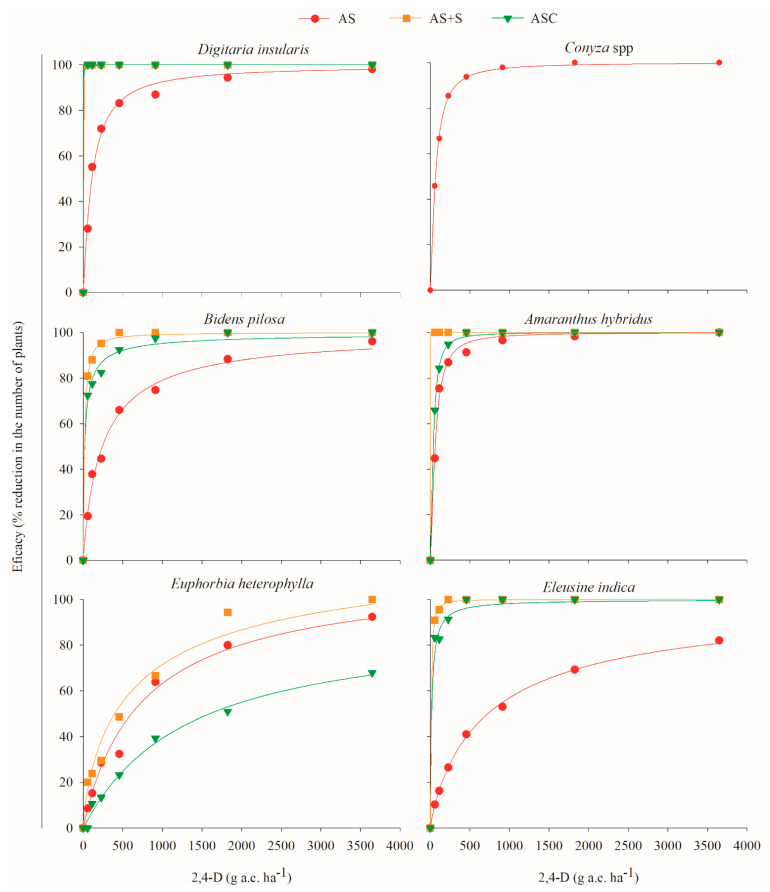
Efficacy of 2,4-D choline salt in the pre-emergence of *Digitaria insularis, Conyza* spp., *Bidens pilosa, Amaranthus hybridus, Euphorbia heterophylla,* and *Eleusine indica* as a function of the application method. AS = application directly on the soil without subsequent straw cover; AS + S = application on the soil followed by straw cover; ASC = application on the straw covering the soil. All were followed by a simulation of 10 mm of rain. The data are expressed as the percentage of the reduction in the number of plants in relation to the treatment with no herbicide applied for each application type.

**Figure 4 plants-11-02800-f004:**
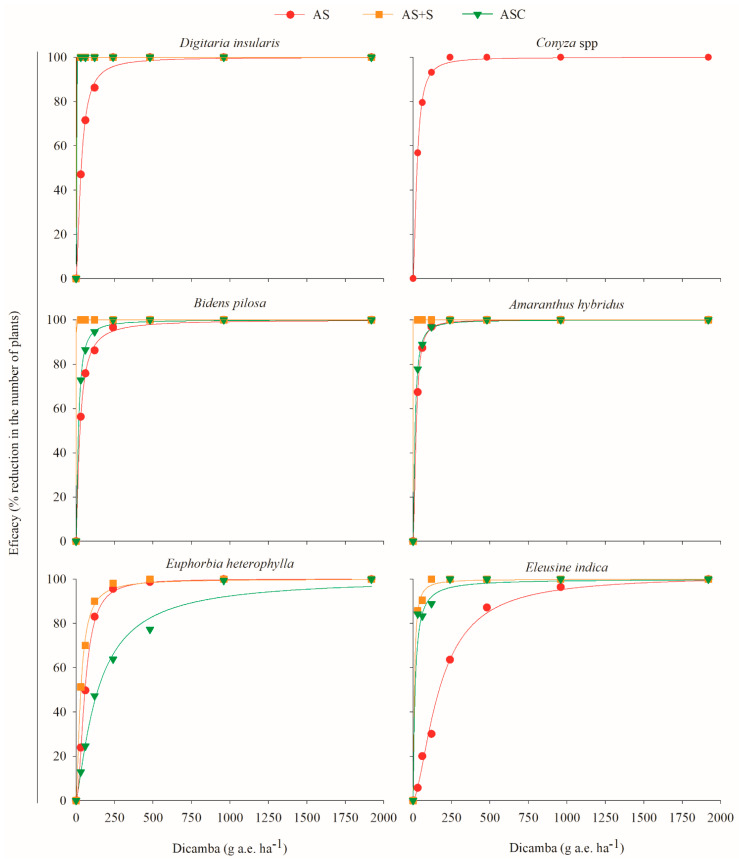
Efficacy of dicamba salt DGA in the pre-emergence of *Digitaria insularis, Conyza* spp., *Bidens pilosa, Amaranthus hybridus, Euphorbia heterophylla*, and *Eleusine indica* as a function of the application method. AS = application of dicamba directly on the soil without subsequent straw cover; AS + S = application of dicamba on the soil followed by straw cover; ASC = application of dicamba on the straw covering the soil. All were followed by a simulation of 10 mm of rain. The data are expressed as the percentage of the reduction in the number of plants in relation to the treatment with no applied herbicide for each application type.

**Table 1 plants-11-02800-t001:** Physicochemical characteristics of the herbicides 2,4-D and dicamba, without considering their salt formulations.

Characteristics	2,4-D	Dicamba
Chemical group	phenoxy-carboxylic acid	benzoic acid
Chemical structure	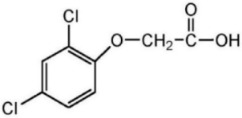	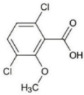
Molecular formula	C_8_H_6_Cl_2_O_3_	C_8_H_6_Cl_2_O_3_
Solubility in water, 20 °C (mg L^−1^)	24,300	250,000
Dissociation constant—pKa at 25 °C	3.4	1.87
Octanol-water partition coefficient at pH 7, 20 °C (Log P)	−0.82	−1.88
Soil adsorption coefficient (Koc) (L kg^−1^)	39.3	1.42
Henry’s law constant at 25 °C (Pa m³ mol⁻¹)	4.0 × 10^−6^	5.06 × 10^−5^
Vapour pressure, 20 °C (mPa)	0.009	1.67
Half-life (days)	4.4	9.62

Source: PPDB: Pesticide Properties Database: University of Hertfordshire [15], Koc–dicamba [16].

**Table 2 plants-11-02800-t002:** Estimates of the parameters of the Mitscherlich model adjusted for the concentrations of 2,4-D and dicamba leached from corn straw (5 t ha^−1^) as a function of the days after application and simulations of different rainfall amounts (mm).

Days without Rain	Model Parameters				F Value
	a	b	c	R²	
2,4-D choline salt					
0	86.3723	0	0.0485	0.9992	2656.9100 **
1	76.4020	0	0.0469	0.9985	1334.6100 **
3	73.3179	0	0.0500	0.9982	1147.8600 **
7	66.7652	0	0.0356	0.9987	1553.2900 **
15	51.7479	0	0.0381	0.9980	1003.4900 **
Dicamba salt DGA					
0	61.0288	0	0.0662	0.9987	11,330.2000 **
1	54.8658	0	0.0531	0.9991	2809.9900 **
3	54.7578	0	0.0495	0.9989	2313.0500 **
7	47.5389	0	0.0351	0.9992	3242.8700 **
15	44.6515	0	0.0356	0.9984	1573.5600 **

Mitscherlich model: Y = a · [1 − 10 (−c·(X + b))]; ** Significant according to the F test at 1% probability.

**Table 3 plants-11-02800-t003:** Estimates of the parameters in the nonlinear model proposed by Streibig ^a^ and CL_80_
^b^ and CL_90_
^c^ values adjusted for the efficacy of the herbicides 2,4-D choline and dicamba DGA salt for different weeds according to the application mode.

Species	Application	Model Parameters	F Value	CL_80_	CL_90_
a	b	X_0_	R^2^
2,4-D choline salt
*Digitaria insularis*	AS	100.8917	1.1257	107.1734	0.9924	30.14 **	367.2140	754.7010
AS + S	100	1.9752	0.0418	0.8347	40.39 **	0.0843	0.1271
ASC	100	1.9752	0.0390	0.8759	56.47 **	0.0787	0.1186
*Conyza* spp.	AS	99.8945	1.3821	65.9158	0.9995	74.47 **	179.7220	323.1610
*Bidens pilosa*	AS	99.6850	0.9410	242.2398	0.8843	65.67 **	1056.9500	2505.1800
AS + S	99.9880	1.1655	17.1909	0.8957	69.50 **	56.4763	113.2500
ASC	99.9188	0.7508	18.5254	0.8035	33.84 **	117.3980	345.7290
*Amaranthus hybridus*	AS	100.301	1.4895	61.6281	0.9421	142.26 **	156.3080	269.4170
AS + S	100.000	1.9465	0.0290	0.9603	193.58 **	0.0591	0.0896
ASC	99.9764	1.6085	38.361	0.9243	98.41 **	90.8216	150.3630
*Euphorbia heterophylla*	AS	96.574	1.2701	657.7352	0.9030	88.09 **	1959.1900	3709.9200
AS + S	91.3852	1.3934	507.0584	0.8608	62.97 **	1371.3200	2454.0800
ASC	101.2237	0.8922	1579.679	0.9922	24.16 **	7470.9000	18,539.900
*Eleusine indica*	AS	99.6114	0.888	732.8578	0.9987	61.69 **	3491.5300	8701.9600
AS + S	99.9991	1.5233	12.8008	0.9997	23.65 **	31.8032	54.1587
ASC	99.9616	0.9884	17.4681	0.9950	8.90 **	71.0185	161.3200
Dicamba salt DGA
*Digitaria insularis*	AS	99.8782	1.5825	32.9172	0.9972	23.85 **	79.0445	131.953
AS + S	100	1.8838	0.0189	1.0000	7.63 **	0.0394	0.0606
ASC	100	1.8953	0.0404	1.0000	6.52 **	0.0839	0.1287
*Conyza* spp.	AS	99.9489	1.6951	25.8267	0.9992	19.90 **	58.5121	94.4084
*Bidens pilosa*	AS	99.9519	1.2979	24.949	0.9984	22.89 **	72.5981	135.6040
AS + S	100	1.96	0.0173	0.9041	75.46 **	0.0350	0.0530
ASC	99.9847	1.4081	15.0433	0.9993	10.42 **	40.2636	71.6183
*Amaranthus hybridus*	AS	99.9881	1.8351	20.2642	0.9108	81.87 **	43.1330	67.1003
AS + S	100	1.9676	0.0187	0.9528	161.61 **	0.0378	0.0512
ASC	99.99	1.4423	12.8027	0.9994	60.16 **	33.4759	58.7372
*Euphorbia heterophylla*	AS	99.2256	2.0074	57.7511	0.9401	128.51 **	115.2070	172.5530
AS + S	99.7938	1.4993	30.4375	0.9971	72.66 **	76.7308	131.7860
ASC	99.3794	1.2855	143.741	0.8483	48.03 **	422.5980	794.133
*Eleusine indica*	AS	97.6066	1.8034	180.5425	0.8749	58.30 **	389.4280	610.5410
AS + S	99.9931	1.3438	8.2821	0.9978	32.84 **	23.2364	42.4862
ASC	99.965	1.1119	13.2203	0.9962	9.18 **	45.9950	95.3784

^a^ Streibig model: y = a/(1 + (x/X_0_)^b^)). ^b,c^ dose of herbicide in g a.e. ha^−1^ required to promote 80 and 90% control. ** Significant according to the F test at 1% probability. AS = application of the herbicide directly on the soil without subsequent straw cover; AS + S = application of the herbicide on the soil followed by straw cover; ASC = application of the herbicide on the straw cover of the soil. All were followed by a simulation of 10 mm of rain.

## Data Availability

Not applicable.

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
