# Peer review of "Dynamics of 2,4-D and Dicamba Applied to Corn Straw and Their Residual Action in Weeds"

_plants, 2022, doi:10.3390/plants11202800_

Round 1

Reviewer 1 Report

The manuscript titled " Dynamics of 2,4-D and dicamba applied to corn straw and their residual action in weeds" is an interesting manuscript that provides new and important data in the field of plant protection. The manuscript is well-structured, and the conclusions are according to the data showed. The abstract is concise. However, are needed minor corrections in order to improve the quality of the manuscript.

I suggest following replacements:

L96 -  greatest replaced with highest

L98 – greater - higher

L101 - greater – higher

L103 greater – higher

and so on ­ greater to be replaced with higher

 L136 – close to- to be replace with up to

L142-144The application of herbicides to the soil, either  by direct application or by movement in the straw (Figure 1), influences the emergence of  mono- and eudicotyledonous weeds –

 I suggest this sentence

L142-144 The herbicides applied to the soil, either by direct application or by movement in the straw (Figure 1), influences the emergence of  mono- and eudicotyledonous weeds

I cannot find references 41, 49, 50 cited in the text of the manuscript.

Author Response

We appreciate the considerations made to the manuscript, which will be considered to improve the final version.
Substitutions for the term heigher were accepted.
The suggestions in lines 136 and 142-144 were also considered.
Citations not found have been added, corrected and updated in the reference list.

Reviewer 2 Report

1.     Title: the title is not clear enough to present the study

2.     Abstract: the abstract is too long, and there is no detailed data presented to support the results

3.     Introduction: there are numerous 2,4-D and dicamba related articles, a more synthetized references review, as the background of this study is needed.

4.     Results: Figure 1, is it better to mark each sub-figures by A, B, C, D and, for example, cite the data of figure as Figure 1A when needed. What are the bars mean, SD or SE or CI?

5.     Results: Figure 2, the histogram (or colum) of Conyza spp is not completely shown, need improve the presentation of the figure

6.     Results: Figure 3 and Figure 4, see comments for Figure 1  

7.     Materials and Methods: where and when was the experiment conducted?

8.     Materials and Methods: Line 523, equation 1 is not complete

Author Response

We appreciate the considerations made to the manuscript, which will be considered to improve the final version.

  1. Title: the title is not clear enough to present the study

We believe that the title portrays the proposal of the work well, being a summary of what was approached in the research. But if you have any suggestions we can look into it.

  1. Abstract: the abstract is too long, and there is no detailed data presented to support the results.

In order not to extend the summary further, we chose to bring the main results, not detailing the data obtained.

  1. Introduction: there are numerous 2,4-D and dicamba related articles, a more synthetized references review, as the background of this study is needed.

We added more information about dynamics and residual action of herbicides. Despite numerous articles related to the two herbicides, few are related to the objective of the work and in order not to make the introduction long, we chose to develop these issues in the discussion.

  1. Results: Figure 1, is it better to mark each sub-figures by A, B, C, D and, for example, cite the data of figure as Figure 1A when needed. What are the bars mean, SD or SE or CI?

The lettering of the sub-figures was considered and added to the manuscript.

The bars represent the confidence interval (p ≤ 0.05).

  1. Results: Figure 2, the histogram (or colum) of Conyza spp is not completely shown, need to improve the presentation of the figure.

As the control for Conyza was 100%, the bar was up to the top line of the chart. To correct, the lines from the top and right side were removed.

  1. Results: Figure 3 and Figure 4, see comments for Figure 1

 In this case, we believe that there would be no need to mark the sub-figures with letters, as each sub-figure has the title at the top, which would already serve as a marker.

  1. Materials and Methods: where and when was the experiment conducted?

The experiments were conducted between 2019 and 2020 in a greenhouse and laboratory at the Nucleus for Advanced Research in Matology - NUPAM, College of Agricultural Sciences, São Paulo State University (Universidade Estadual Paulista “Júlio de Mesquita Filho” UNESP), Botucatu, SP, Brazil

  1. Materials and Methods: Line 523, equation 1 is not complete

The equation has been corrected. y = a[1-10(-c(x+b))]